# User Perceptions of ROTEM-Guided Haemostatic Resuscitation: A Mixed Qualitative–Quantitative Study

**DOI:** 10.3390/bioengineering10030386

**Published:** 2023-03-21

**Authors:** Greta Gasciauskaite, Amos Malorgio, Clara Castellucci, Alexandra Budowski, Giovanna Schweiger, Michaela Kolbe, Bastian Grande, Christoph B. Noethiger, Donat R. Spahn, Tadzio R. Roche, David W. Tscholl, Samira Akbas

**Affiliations:** 1Institute of Anaesthesiology, University Hospital Zurich, Frauenklinikstrasse 10, 8091 Zurich, Switzerland; 2Simulation Center, University Hospital Zurich, Gloriastrasse 19, 8091 Zurich, Switzerland

**Keywords:** haemostasis, coagulation management, point-of-care, viscoelastic test, rotational thromboelastometry, user-centred design, situation awareness

## Abstract

Viscoelastic point-of-care haemostatic resuscitation methods, such as ROTEM or TEG, are crucial in deciding on time-efficient personalised coagulation interventions. International transfusion guidelines emphasise increased patient safety and reduced treatment costs. We analysed care providers’ perceptions of ROTEM to identify perceived strengths and areas for improvement. We conducted a single-centre, mixed qualitative–quantitative study consisting of interviews followed by an online survey. Using a template approach, we first identified themes in the responses given by care providers about ROTEM. Later, the participants rated six statements based on the identified themes on five-point Likert scales in an online questionnaire. Seventy-seven participants were interviewed, and 52 completed the online survey. By analysing user perceptions, we identified ten themes. The most common positive theme was “high accuracy”. The most common negative theme was “need for training”. In the online survey, 94% of participants agreed that monitoring the real-time ROTEM temograms helps to initiate targeted treatment more quickly and 81% agreed that recurrent ROTEM training would be beneficial. Anaesthesia care providers found ROTEM to be accurate and quickly available to support decision-making in dynamic and complex haemostatic situations. However, clinicians identified that interpreting ROTEM is a complex and cognitively demanding task that requires significant training needs.

## 1. Introduction

Severe bleeding and coagulopathy are critical conditions associated with surgical procedures, obstetrics and trauma. Critical haemorrhage is linked to high mortality rates [1,2,3,4]. Prompt and precise diagnosis of the underlying coagulopathy and their aetiologies is of utmost importance in enabling targeted life-saving treatment [1,3]. Standard coagulation tests often take more than an hour until results are available and are, therefore, not an optimal approach when urgent therapeutic action is required [2,5].

Viscoelastic haemostatic assays, such as rotational thromboelastometry (ROTEM^®^), deliver real-time graphical and numerical assessment of global clot formation and dissolution and are particularly important in making time-efficient decisions about haemostatic interventions [1,6]. Viscoelastic haemostatic assays provide information on haemostatic efficacy based on the surrogate endpoint of maximum clot firmness. They enable the evaluation of additional aspects, such as clot formation kinetics and fibrin-platelet interactions [7]. The technology detects disorders such as hyperfibrinolysis, coagulation factor deficiency and thrombocytopenia, providing first results within ten minutes of blood sampling. Such early detection of coagulation disorders can improve patient outcomes, particularly in trauma patients with major bleeding [1,8,9]. Viscoelastic haemostatic assays are also essential in many other medical fields: prior studies showed the benefits of using ROTEM in neuro [10], cardiac [11,12], transplant [13], burn [14] and paediatric [1] surgery.

Viscoelastic haemostatic tests are also paramount in haematological diseases, for instance, in the perioperative management and diagnostics of congenital quantitative fibrinogen disorders such as hypofibrinogenemia or afibrinogenemia [15,16].

Several European and North American transfusion recommendations [17,18,19] emphasise the importance and benefits of viscoelastic haemostasis testing, including the reduced need for transfusion, fewer perioperative complications, shorter hospital stays and lower treatment costs.

Despite these significant benefits and widespread clinical use, accurate analysis can be challenging even for experienced care providers. It is, therefore, necessary to consider how clinicians working with this technology can be supported. One of the solutions for significantly improving patient care is the application of artificial intelligence and machine learning [20,21,22,23]. Situation awareness and user-centred design principles also play an essential role in facilitating decision-making in complex clinical situations [24,25].

The aim of this study was to learn more about the perceptions of anaesthetists working with ROTEM, with a view to identifying and specifying the positive user perceptions regarding the technology and recognising the potential for future improvements.

## 2. Materials and Methods

### 2.1. Approval and Consent

The Cantonal Ethics Committee of the Canton of Zurich, Switzerland, reviewed the study protocol and issued a declaration of no objection (Business Management System for Ethics Committees Number Req-2021-01112). However, each participant gave informed consent to use his or her data for research purposes. Participation was voluntary and without financial compensation.

### 2.2. Study Design

This study is a researcher-initiated, single-centre, mixed qualitative–quantitative study investigating anaesthetists’ perceptions of ROTEM-guided haemostatic resuscitation. The study was performed at the University Hospital Zurich, Institute of Anaesthesiology, Switzerland, over three consecutive weeks in January and February 2022.

We first interviewed anaesthesia staff, including staff physicians, residents, and nurses, about the positive and negative aspects of ROTEM immediately after they had participated in a high-fidelity simulation study of perioperative bleeding scenarios.

In the second step, a group of anaesthetists and nurses completed an anonymous online survey, which we emailed to all staff physicians, residents and nurses at the Institute of Anaesthesiology, to quantitatively rate statements we generated based on themes identified in the interviews.

Our study design accepted the overlap between the interview participants and the online survey participants, as the online survey was designed to quantify the interview responses and not to validate the statements per se.

In the study centre, ROTEM is the standard of care for the management of acute haemorrhage, so all providers were familiar with the method before participating in the simulation study.

### 2.3. Previous High-Fidelity Simulation Study

In the high-fidelity simulation study, anaesthesia teams, consisting of a staff anaesthesiologist, a resident and an anaesthesia nurse, performed a total of 60 high-fidelity simulations (4 per team) of perioperative bleeding using standard ROTEM temograms or an experimental, user-centred visualisation technology—Visual Clot. Visual Clot is a new visualisation technology—an animated 3D blood clot that illustrates ROTEM parameters in a user-centred way [26,27,28].

Before the simulations began, participants were given a standardised presentation of about 10 min duration, which reviewed ROTEM and introduced Visual Clot. They were invited to ask questions to clarify any uncertainties. Figure 1 shows an example of a ROTEM, and Visual Clot printout as used in the simulation study.

### 2.4. Participant Interviews and Online Survey

Both before the interviews and in the online surveys, participants answered a series of non-identifying demographic questions and questions about their anaesthesia and ROTEM experience.

The interviews took place at the end of the simulation sessions. While participants freely verbalized their impressions in a distraction-free environment, the data collectors took field notes. The only guidance given to the participants prior to the interviews was to formulate their positive and negative perceptions of ROTEM. At the end of the interviews, the collected responses were shown to the participants, who were encouraged to make final adjustments.

In addition, to summarise and consolidate the findings from the interviews, we generated six statements based on the themes that emerged from the participants’ responses. Such a mixed qualitative–quantitative approach is supported by literature [29,30,31]. For the online survey, an email invitation was sent to all care providers in the Institute of Anaesthesiology. This survey asked participants to rate the statements generated from the interview responses on Likert scales.

### 2.5. Outcomes and Statistical Analyses

#### 2.5.1. Part I: Participant Interviews

Collected answers were translated from the original German to English using an online translation system, DeepL (DeepL GmbH, Cologne, Germany). All translated field notes are provided in Appendix A. In order to identify the main themes that emerged from the participants’ responses, we used a template approach and generated a coding tree [32]. According to this coding template, statements were allocated to themes (Figure 2).

Using the word count function, we identified the most frequently mentioned terms in positive and negative responses, uniting word groups with the same word root and excluding the commonly used filler words such as “the” or “and”, which helped us to recognise similar expressions. The word clouds created using WordArt.com (Wordart GmbH) (accessed on 30 December 2022) provide visualisations of the repeatedly mentioned terms. The more commonly used words are shown as more prominent and appear more frequently in different sizes than those mentioned less often. Figure 3 shows the most frequently used positive concepts, and Figure 4 displays the most commonly mentioned expressions regarding ROTEM disadvantages.

Three anaesthesiology residents and study authors GG, GS and SA evaluated the statements independently of each other using a coding template (Figure 2). Interrater reliability was estimated to examine the consistency of application of the coding template. In case of disagreement between three examiners after multiple data coding, a final decision was made in a joint discussion.

#### 2.5.2. Part II: Online Survey

In the online survey, we defined six statements based on the previously identified themes and asked participants to rate them on five-point Likert scales, with responses ranging from “strongly disagree” to “strongly agree”. The questionnaire created using Google Forms (Alphabet Inc., Mountain View, CA, USA) was sent via E-Mail to all senior doctors, residents and anaesthesia nurses at the University Hospital of Zurich. In the survey invitation, we informed participants that the questionnaire takes about two minutes to complete and participation is voluntary. The translated survey invitation announcement is provided in Appendix A. Participants were able to review and finalize their answers before completion. No personal identifying information was collected during the survey. The data collection was finished 2 weeks after the survey was sent.

### 2.6. Statistical Analysis

For the management of the interview data analysis and generation of the figures, we used Microsoft Word and Excel (Microsoft Corp., Redmond, WA, USA). We report the number of statements and their percentage distribution in the identified themes.

Cohen’s kappa was calculated using R, version 4.0.5 (R Foundation for Statistical Computing, Vienna, Austria), to define the interrater reliability of the coding template.

For the online survey analysis, we calculated each statement’s median and interquartile range. Statistical differences between the median and neutral answers were determined using the Wilcoxon signed-rank test. *p* < 0.05 was interpreted as indicating statistical significance.

## 3. Results

### 3.1. Study and Participant Characteristics

Table 1 provides detailed information on the study and participants.

In the interviews, residents and nurses accounted for 90 percent of participants in approximately equal proportions. The least experienced participant had less than one year of experience in anaesthesia. The most experienced had 33 years. Experience with ROTEM varied from no experience to more than 100 ROTEMs per year.

In the online questionnaire, in contrast to the interviews, most participants were senior physicians. The median anaesthesia experience did not differ from the interviewed group, but the average number of ROTEM interpretations was higher in this group.

### 3.2. Part I: Qualitative Analysis of Interview Answers

#### 3.2.1. Word Count Analysis

The word count analysis identified these four most commonly mentioned words participants used to describe the advantages of ROTEM: quick/quickly/quicker; fast/faster (18/77 participants, 23.4%), visible/visual/visualization (13/77 participants, 16.9%), information (11/77 participants, 14.3%), and quantification/quantitative (9/77 participants 11.9%).

We also identified the following three words and word combinations commonly used by participants to describe the limitations and potential for ROTEM improvement: interpretation (18/77 participants, 23.4%), experience (12/77 participants, 15.6%), and complex/complicated (9/77 participants, 11.7%).

Figure 3 and Figure 4 provide a visual representation of the most commonly occurring positive and negative words.

#### 3.2.2. Coding Template

Figure 2 demonstrates the generated coding pattern, including two main domains and ten themes. The interrater reliability was 0.799 (95% confidence interval from 0.776 to 0.822), indicating very substantial agreement.

#### 3.2.3. Statements about ROTEM: Major Topics and Subthemes

A total of 288 statements were collected during the interviews, 148/288, 51.4% positive and 140/288, 48.6% negative ones.

As shown in Figure 2, we divided the positive and negative statements into ten themes: five topics in the positive and five in the negative group.

Four comments out of 148 (2.7%) in the positive domain have not been assigned to any theme; they are non-codable. While in the group of negative perceptions, 11 statements out of 140 (7.86%) were non-codable.

Table 2 displays the major domains and themes with participant counts, percentages and examples. Figure 5 shows the percentage distribution among the main topics and themes.

#### 3.2.4. Themes

Positive perceptions about ROTEM

Positive Design Features

Concerning positive design features, the participants emphasised the real-time monitoring of the temograms as a simplifying factor in interpreting the results. Participant #17 outlined that “graphics are easy to interpret” while participant #21 pointed out that “visual representation, not only numbers” and a “compact presentation” of both make information easier to perceive. Another essential factor that dominated the positive insights describing the design was a clear structure. Participant #55 described ROTEM as “well-structured”, and participant #44 specified that the “categorically divided (intrinsic/extrinsic)” model contributes to the easier interpretation of results. Graphic design features also increase ROTEMs attractiveness: “not too many colours” (participant #51) and “visual analysis (with colour and shape)”, as participant #19 said.

Positive Usability Features

Concerning positive usability features, the point-of-care diagnostic was repeatedly underlined (participants #7, 9, 35, 74), which also contributes to a “beneficial and efficient (usage) in the acute situation” (participant #6). User-friendliness of ROTEM Sigma was uplifted compared to ROTEM Delta: “ROTEM Sigma is easy to handle, no more pipetting required” (participant #10).

Quick Availability of Results

Faster access to results compared to standard coagulation tests (Participants #19, 33) enables a much quicker response to disorders and simplifies work (Participant #6). Quick availability of results was also an essential point in discussing the usage of ROTEM Sigma compared to ROTEM Delta. It was mentioned that working with ROTEM Sigma produces results much faster (Participant #11) because of positive usability features, as discussed above.

Proven coagulation management tool

Underlying this theme, ROTEM was emphasised as the “standard you are used to” (participant #54), a “familiar technique”(participant #67), which has become a “habit” (participant #32) over many “years of experience” (participant #8).

High Accuracy

The quantitative, numerical information provided by ROTEM was repeatedly mentioned as a factor contributing to the accuracy: “exact quantitative values” (Participant #4), “more information, quantification” (Participant #61), “quantification available” (Participant #71). The importance of the real-time monitoring of the temograms as a qualitative characteristic of ROTEM was also mentioned as a factor that increases accuracy.

Negative perceptions about ROTEM

Negative Design Features

As a potential for improvement in discussing design features, participant #52 pointed out that “it would be helpful if it would blink when something is wrong”, while participant #37 stated that “It might be beneficial if values were marked red, if out of normal range”, underlining the wish for an additional alarming system layer. Another design feature that users have mentioned as a negative one was related to the graphics of the display: “Numbers are shown too small” (participant #37), “numbers are small in relation to the image” (participant #55).

Usability Problems

Concerning negative usability features, participants especially outlined that interpreting information provided by ROTEM is a complex and cognitively very demanding task, particularly in emergencies (participants #15, 16, 21, 41, 45, 49).

Interference Susceptibility

The sensitivity of the device and associated technical interferences have been mentioned in the comments attributed to this topic. Participant #23 stated, “Even if you take a wrong blood collection tube, there are problems, for example, those with less blood volume”. 

Incompleteness

The fact that ROTEM does not fully represent the coagulation status was emphasised multiple times (participants #3, 9, 19, 26). Participant #26 outlined this as a significant limitation, especially in acute cases.

Need for training

The most comments (54/288, 19.8%) were made on this theme, emphasizing that ROTEM must be learned extensively (participant #5) and that “with a lack of knowledge, the ROTEM cannot be interpreted” (participant #18). It makes using ROTEM “not so easy for beginners” (participant #24), and “it takes time to master it” (participant #36).

### 3.3. Part II: Analysis of Statements Evaluated in the Online Survey

The detailed evaluation of the statements rated in the online survey is illustrated in Figure 6. Except for statement #5, the sample medians differed statistically significantly from neutral (*p* < 0.05).

## 4. Discussion

### 4.1. Principal Findings

This mixed qualitative–quantitative study analysed anaesthetists’ perceptions of ROTEM-guided haemostatic resuscitation. ROTEM is an alternative to standard coagulation tests for personalised decision-making in time-critical coagulation interventions. Another commonly used approach, supporting care providers when making decisions about personalised haemostatic interventions, is thromboelastogram (TEG ^®^). This technology, as ROTEM, allows the viscoelastic properties of clot formation and dissolution to be evaluated in real-time. ROTEM is a modern alteration of TEG [6]. We analysed user perceptions to highlight aspects where ROTEM is perceived positively and to identify areas where care providers see room for improvement that could help future development.

The main findings show that the accuracy of information provided by ROTEM, due to the combination of numerical data presentation and temograms, is the most important positive feature from the users’ perspective. The ROTEM design, which allows real-time monitoring of coagulation status, was also very well received. The rapid availability of results was repeatedly mentioned by participants in our study as a feature that makes the use of ROTEM more attractive. According to these results, participants’ perceptions were consistent with ROTEMs intended purpose of providing high diagnostic accuracy and a technology design that enables real-time haemostatic monitoring to quickly initiate personalized treatment [6,33]. The study shows that users perceive ROTEM as a standard haemostatic resuscitation method and that, as participant #32 said, its use is perceived as a habit.

The main criticism was focused on the extensive cognitive resources required to interpret ROTEM results. Participants described non-intuitiveness of the result presentation as a challenge, particularly in emergencies and for inexperienced users. The high demand for training was also reflected in the online survey, where 81% of respondents agreed that they would find recurrent training beneficial. In emergencies, caregivers are cognitively challenged by emotional stress and cognitive load, the combination of which can overwhelm cognitive resources and lead to a decline in performance [34,35,36]. Situation awareness and user-centred design are concepts that can help health professionals make clinical decisions in dynamic situations [25,37,38].

The concept of situation awareness illustrates why it is important to minimise the cognitive load required to understand the presentation of results from diagnostic tools. Situation awareness—first defined by Endsley in the late 1980s—consists of three progressive levels: I. perceiving the relevant elements in a situation, II. understanding their meaning, and III. projecting their status into the near future. Only when the situation is correctly perceived and understood can the right decisions be made. Free mental capacity is necessary to build optimal situation awareness [24]. In particular, in stressful situations, this should not be compromised by complicated data interpretation [39]. One way to increase situation awareness is through user-centred design. It helps reduce the cognitive burden and assists users in decision-making. User-centred design aims to integrate information in ways that are adapted to the users’ tasks, goals and needs, to create optimal human-machine interaction. It leads to a reduction in errors and an improvement in productivity, acceptance and user satisfaction [24]. There are some examples of such technologies based on the principles of situation awareness and user-centred design [39,40,41]. To help healthcare providers interpret the results of viscoelastic coagulation tests, Visual Clot, a new technology for alternative presentation of thromboelastometry data has been developed. In this technology, an algorithm takes parameters from ROTEM and creates a visual representation in the form of a 3D animated blood clot. A computer-based simulation study revealed that the perceived cognitive workload was lower and perceived diagnostic confidence was higher when working with Visual Clot than when using conventional ROTEM temograms [26,28]. These findings are an invitation to rethink user needs in order to develop more intuitive technologies that reduce cognitive load and help streamline the decision-making process.

Artificial intelligence and machine learning are other essential facets inseparable from today’s medicine and the facilitation of decision-making. Its applications found their way into clinical routine, significantly improving patient care [21,23]. Multiple studies have explored the potential use of artificial intelligence in many areas of medicine, such as cardiology [21], neurology [20], oncology [22], haematology [42], nephrology [43], gastroenterology, hepatology, orthopaedics and rheumatology [21]. The findings hold great promise for revolutionising clinical care, not only in terms of better diagnostic and therapeutic options for patients but also by facilitating decision-making and reducing cognitive load for clinicians.

This study revealed a wide range of user perceptions of ROTEM and made us aware of their needs. The comprehensive results could help us to design the technology around the users’ tasks, skills and illustrate the need to simplify decision-making.

### 4.2. Strengths and Limitations

This study has a number of strengths and limitations. The interview part of the study has the inherent limitations of qualitative research. A qualitative analysis provides a full, detailed description without trying to assign frequencies to the features identified in the data. Rare and more common phenomena receive equal attention. The results of qualitative analysis cannot be extrapolated to larger populations with the same certainty as quantitative results because the findings are not tested for statistical significance [44]. However, following up the qualitative research with a quantitative online survey helped to provide greater insight into the significance of the key themes identified. As part of the simulation study, the interviewees received an introduction to Visual Clot and a review of ROTEM. Seeing the two technologies side by side shortly before the interviews, with their various advantages and disadvantages, may have influenced the participants’ responses.

Interviewees were not selected randomly, but according to their scheduling in the high-fidelity simulation study. However, this selection was quasi-random and independent of the study. In addition, the second part of the study, the online survey was sent to all doctors and nurses in the Institute. The high participation in the online survey and the balanced gender and occupation sample (doctors and nurses) further reduced possible selection bias.

Finally, this is a single-centre study conducted in a university hospital with a high standard of care in Europe. User opinions may vary across diverse clinical settings in different parts of the world.

## 5. Conclusions

To our knowledge, this is the first study to investigate the opinions of experienced care providers about ROTEM. The aim of this study was to raise awareness of what anaesthetists perceive to be the strengths and weaknesses of ROTEM-guided haemostatic resuscitation. The study shows that users positively perceive this technology’s design, usability features, quick availability of results, and high accuracy. The high accuracy was the most frequently mentioned positive feature of ROTEM, enabling rapid initiation of personalized treatment. ROTEM was generally described as user-friendly and well-designed.

However, the participants underlined that interpretation of ROTEM requires significant cognitive resources—extensive theoretical knowledge and a lot of practical experience. These findings encourage us to explore a more user-oriented and situation awareness-based presentation of the ROTEM results.

## Figures and Tables

**Figure 1 bioengineering-10-00386-f001:**
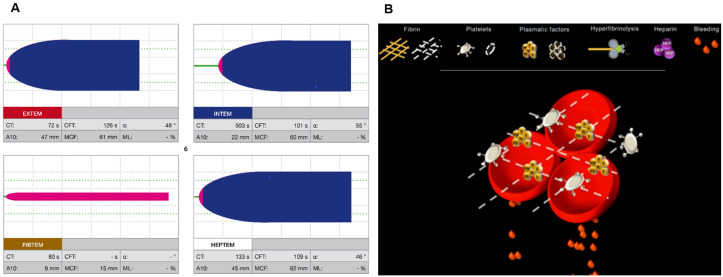
(**A**): A conventional ROTEM result, showing a heparin effect. (**B**): A Visual Clot representation. The fibrin in the clot is shown as a dashed line, indicating its absence.

**Figure 2 bioengineering-10-00386-f002:**
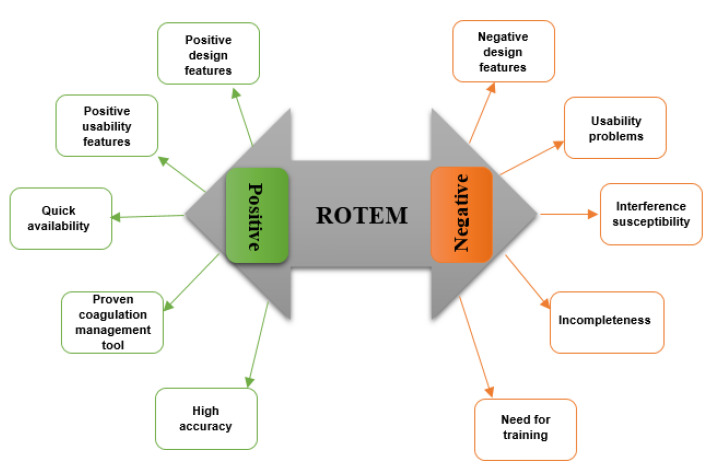
A coding template, demonstrating the major themes describing positive and negative user perceptions.

**Figure 3 bioengineering-10-00386-f003:**
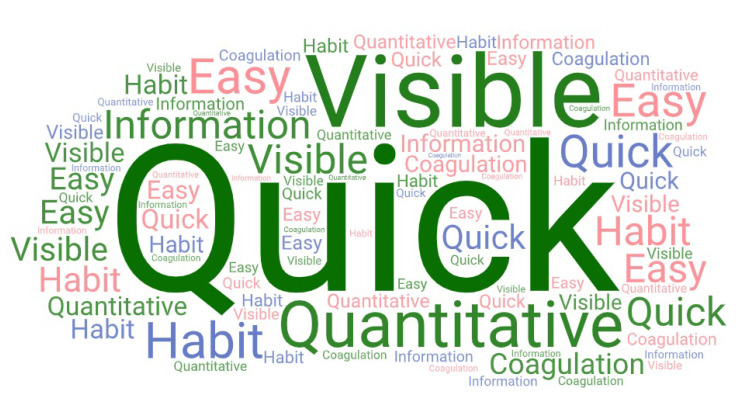
A quantitative graphic representation of the most frequently used positive terms describing ROTEM as a word cloud. More commonly mentioned expressions have larger front sizes. Common filler words (and, to, the, etc.) are not shown. The graphic was created using WordArt.com (accessed on 30 December 2022).

**Figure 4 bioengineering-10-00386-f004:**
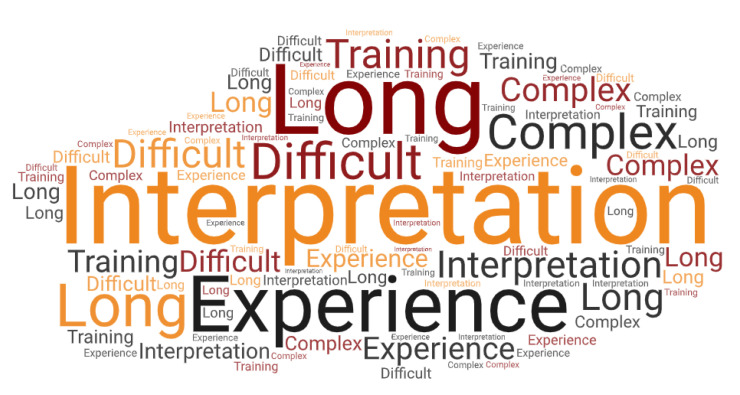
A tag cloud quantitatively demonstrating the most common words describing the negative user perceptions of ROTEM. The size of the words corresponds to their frequency of occurrence. Filler words (and, to, the, etc.) are not displayed. The word cloud was created using WordArt.com (accessed on 30 December 2022).

**Figure 5 bioengineering-10-00386-f005:**
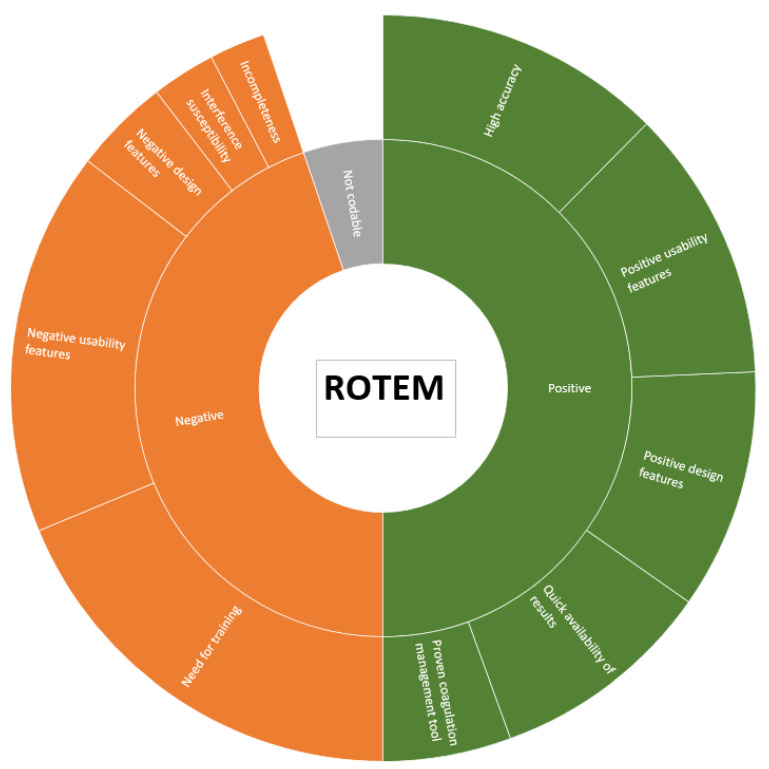
Sunburst diagram reflecting the percentage distribution among the main topics and sub-themes.

**Figure 6 bioengineering-10-00386-f006:**
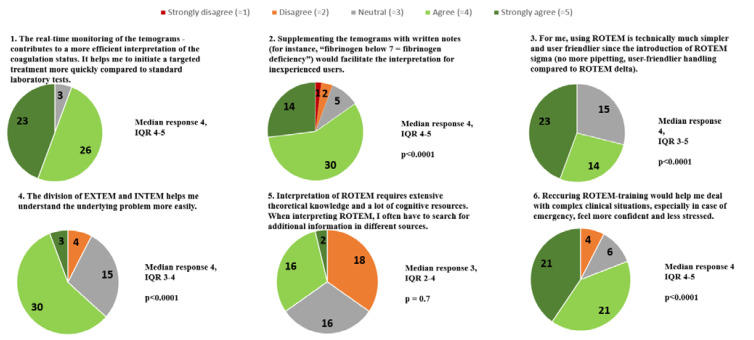
Pie charts presenting online survey results with the number of participants who chose a particular category (*n* = 52). The results are presented as medians and interquartile ranges (IQR). *p*-values are provided to indicate a statistically significant difference between the median of the sample and the neutral value.

**Table 1 bioengineering-10-00386-t001:** Study and participant characteristics.

Participant Characteristics	
Interview Participants (n = 77)	
Female participants, n (%)	46 (59.7)
Senior physicians, n (%)	8 (10.4)
Resident physicians, n (%)	35 (45.5)
Anaesthesiology nurses, n (%)	34 (44.2)
Anaesthesia experience in years, median (IQR)	8 (3–10)
Number of ROTEM interpretations per year, median (IQR)	26 (5–41)
Online survey participants (n = 52)	
Senior physicians, n (%)	20 (38.5)
Resident physicians, n (%)	17 (32.7)
Anaesthesiology nurses, n (%)	15 (28.8)
Anaesthesia experience in years, median (IQR)	8 (4–10)
Number of ROTEM interpretations per year, median (IQR)	44 (7.75–50)

**Table 2 bioengineering-10-00386-t002:** The major topics with statement count, percentages and examples.

Major Topics and Subthemes	Examples
Positive Perceptions about ROTEM (148/288, 51.4%)	
Positive design features (30/288, 11%)	Visual. (#30) Time course visible. (#31) Categorically divided (intrinsic/extrinsic)—clearly structured. (#44))
Positive usability features (34/288, 12.5%)	Good overview of coagulation disorders. (#68) Good that you no longer have to pipette. (#70) Quantifying what is missing and to what extent. (#24)
Quick availability of results (28/288, 10.3%)	Can draw a conclusion quickly. (#2) Simplifies work because it gives a quick result and does not have to be sent to the lab. (#6) A lot of information can be obtained in a fast time. (#36)
Proven coagulation management tool (16/288, 2.9%)	It is known and one is used to it. (#14) Universally recognized. (#29) Already imprinted. (#33)
High accuracy (36/288, 13.2%)	Many details and information to extract. (#43) Can be balanced more accurately (4 g fibrinogen instead of only 2 g). (#54) You can see the quantitative, how much is missing. (#53)
Negative perceptions about ROTEM (140/288 48.6%)	
Negative design features (12/288, 4.4%)	Too many numbers, too much red. (#8) Numbers are too small, especially under stress and on night duty. (#17) Not clearly arranged. (#29)
Usability problems (48/288, 17.6%)	Relatively long duration for consideration, although it is used in critical situations. (#7) Complex. (#21) Too difficult to interpret (also because of numbers). (#39)
Interference susceptibility (8/288, 2.9%)	Very sensitive device, so that there are still disturbances. (#23) Unclear why it does not work again and again. (#40) If it is not accepted, in case of measurement error, the whole blood collection tube is automatically gone. (#40)
Incompleteness (7/288, 2.6%)	Limitation of imaging because not all clotting problems are detected. (#9) Does not capture many things. (#3) Too much information that is not evaluated (especially in acute cases). (#26)
Need for training (54/288, 19.8%)	Is not intuitive. (#20) Needs a lot of experience. (#29) It takes time to master it. (#36)
Non-codable	
	ROTEM does not work on all PCs. (#40) Not well-founded in the education. (#70) Glasses must be put on. (#77)

## Data Availability

The datasets used and/or analysed during the current study are available from the corresponding author on reasonable request.

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
