# Peer review of "User Perceptions of ROTEM-Guided Haemostatic Resuscitation: A Mixed Qualitative–Quantitative Study"

_bioengineering, 2023, doi:10.3390/bioengineering10030386_

Round 1

Reviewer 1 Report

The authors of the manuscript focused quantitative and qualitative study of viscoelastic point-of-care in hemostatic resuscitation.  The authors have come up with an interesting original data that is very important for medical research. Rotational thromboelastometry and thromboelastography is a holistic blood coagulation assay.  Rotational thromboelastometry is a viscoelastic hemostatic assay that has been used in emergencies (trauma and obstetrics), and surgical procedures (cardiac surgery and liver transplants). The manuscript is well structured. Some parts of the manuscript need to be corrected and supplemented in order for this manuscript to be published.

Page 1, lines 41-44: Global hemostatic assays provide information on the hemostatic efficacy based on the surrogate end point of maximum clot firmness and allow the evaluation of additional aspects such as the kinetics of clot formation, fibrin – platelet interactions, and the rate of fibrinolysis. These important facts should be mentioned and at the same time cite the manuscript for which it was published: ,,Semin Thromb Hemost. 2016 Jun;42(4):356-65. doi: 10.1055/s-0036-1571340“

Page 2 , lines 49-50: The following facts need to be supplemented in this sectionIn addition to the mentioned medical fields, we can use ROTEM in hematological diseases – in perioperative management and diagnosis of congenital quantitative disorders of fibrinogen. It would be appropriate to cite these manuscripts:: Thromb Res. 2020 Apr;188:1-4. doi: 10.1016/j.thromres.2020.01.024., and  J Thromb Thrombolysis. 2020 Jul;50(1):233-236. doi: 10.1007/s11239-019-01991-x.

Tables and figures  in the text are very clearly written.

I have to say that with these 35 references. Most of the references are from the last 5 years.

Author Response

Dear Prof. Guiseppi-Elie,
Dear Reviewers,
Dear Ms. Kruk,

Thank you for your kind evaluation of our manuscript. We thank the referees for their valuable input, which helped to improve our work. Subsequently, we address all of their comments point by point. You will find it attached in a pdf file.We look forward to a positive re-evaluation of our revised manuscript.

On behalf of all authors with best regards,
Greta Gasciauskaite

Reviewer 2 Report

The paper analyzes a viscoelastic point-of-care haemostatic resuscitation methods (ROTEM) for decision making about personalised coagulation interventions. The paper is well focused on the proposed research topic.  A minor revision is required.

Strengths: the experimental actions are accurately described.  

Points of weakness: the discussion of the method is weak. 

Actions to do:

According to the weaknesses, I suggest to improve the paper by answering to these points:

·       Please add more information about other possible approach suitable for personalized interventions.     

·       Please provide at least one procedure describing a personalized intervention;

·       Please provide more details about tag cloud approach (Fig. 3 , Fig. 4);

·       Further references should be added in the introduction section about (introduction section), such as:

Artificial Intelligence supporting in general clinical classification

https://doi.org/10.3390/diagnostics12082003

doi: 10.1109/MeMeA49120.2020.9137224

https://doi.org/10.3390/jcm11082265

https://doi.org/10.3390/healthcare11010080

Process mining (improvements of the decision making process using artificial intelligence in processes concerning interventions and procedures as for other application fields)

https://doi.org/10.3390/healthcare11020207

https://doi.org/10.3390/s22228677

https://doi.org/10.3390/app12052677

Minor remarks:

Fig. 1 must be more clear (higher resolution) and correct some English grammar errors.

Author Response

Dear Prof. Guiseppi-Elie,
Dear Reviewers,
Dear Ms. Kruk,

Thank you for your kind evaluation of our manuscript. We thank the referees for their valuable input, which helped to improve our work. Subsequently, we address all of their comments point by point. You will find it attached in a pdf file. We look forward to a positive re-evaluation of our revised manuscript.

On behalf of all authors with best regards,
Greta Gasciauskaite

Round 2

Reviewer 1 Report

The presented manuscript has been corrected in response to the suggestions. The authors have followed the recommendations of the reviewer. After the revision, the provided data and addition of the results became more clear. I would like to thank the authors for resubmitting the manuscript and explaining the obscure points from the previous version.